# A Survey on Perceptions of the Direction of Korean Medicine Education and National Licensing Examination

**DOI:** 10.3390/healthcare11121685

**Published:** 2023-06-08

**Authors:** Han-Byul Cho, Won-Suk Sung, Jiseong Hong, Yeonseok Kang, Eun-Jung Kim

**Affiliations:** 1Department of Neuropsychiatry, Graduate School, College of Korean Medicine, Dongguk University, Seoul 04620, Republic of Korea; 2Department of Acupuncture & Moxibustion, Dongguk University Bundang Oriental Hospital, Seongnam-si 13601, Republic of Korea; 3Teaching & Learning, 7 Days Inc., Seoul 06247, Republic of Korea; 4Department of Medical History, College of Korean Medicine, Wonkwang University, Iksan-si 54538, Republic of Korea

**Keywords:** Korean medicine education, national licensing examination for Korean medicine doctors, clinical occupational competency, survey

## Abstract

Recent changes in medical education and assessment led to a focus on occupational competency, and this study investigated the perceptions of Korean medicine doctors (KMDs) on the national licensing examination for KMDs (NLE-KMD). The survey aimed to understand KMDs’ recognition of the current situation, items to improve, and items to emphasize in the future. We conducted the web-based survey from 22 February to 4 March 2022, and 1244 among 23,338 KMDs answered voluntarily. Through this study, we found the importance of competency-related clinical practice and Korean standard classification of disease (KCD), and the presence of a generation gap. KMDs considered clinical practice (clinical tasks and clinical work performance) and the item related to the KCD important. They valued (1) the focus on KCD diseases that are frequently seen in clinical practice and (2) the readjustment and introduction of the clinical skills test. They also emphasized KCD-related knowledge and skills for the assessment and diagnosis of KCD diseases, especially those frequently treated at primary healthcare institutes. We confirmed the generation gap in the subgroup analysis according to the license acquisition period, and the ≤5-year group emphasized clinical practice and the KCD, while the >5-year group stressed traditional KM theory and clinical practice guidelines. These findings could be used to develop the NLE-KMD by setting the direction of Korean medicine education and guiding further research from other perspectives.

## 1. Introduction

Republic of Korea, China, and Japan have their own types of traditional medicine, and each country has its own system of education [1,2]. Mainland China has two education courses. One consists of a five-year bachelor’s degree and one year of clinical training, while the other consists of a seven-year master’s degree only. After completing these courses, students are eligible to select and obtain the following licenses: traditional Chinese medicine (TCM), Western medicine, and TCM/Western medicine. In contrast, Japan does not have its own license for Kampo medicine (the name of traditional Japan medicine). Kampo medicine training is provided as an optional and additional education after graduation from medical university, and the Japan Society for Oriental Medicine grants students who completed the training the authority to perform Kampo treatment.

Republic of Korea has two education pathways (six-year program and four-year postbaccalaureate program), and (prospective) graduates are eligible to apply for the national licensing examination for Korean medicine doctors (NLE-KMD). The Korea Health Personnel Licensing Examination Institute (KHPLEI) oversees NLE-KMD in Korea. Based on the KHPLEI, a Korean medicine doctor (KMD) is responsible for diagnosing the patients based on the theory of Korean medicine (KM), conducting KM treatments (e.g., acupuncture, herbal medicine), and overseeing rehabilitation and public health prevention [3].

The KHPLEI established the current frame of the NLE-KMD in 1990 based on the following KM college subjects: internal medicine 1 and 2 (including Sasang constitutional medicine (SCM)), acupuncture and moxibustion, medical law, dermatology and surgery, neuropsychiatry, ophthalmology and otorhinolaryngology, gynecology, pediatrics, preventive medicine, physiology, and herbology [4]. The KHPLEI continuously implemented the subject-centered paper examination system without significant changes, with pass criteria of at least 60% for the total score and at least 40% for each subject [4].

However, the current medical education and examination system is being replaced with a clinical occupational competency-centered approach [5]. The Korean medical licensing examination (KMLE) went through several revisions, and the clinical skills test was introduced in 2009. This test contains history taking, physical examination, physician and patient interaction, attitude toward patients, and basic technical skills, and it aims to evaluate doctors’ clinical competence [6]. These changes are progressing domestically and internationally in several fields, including medicine, dentistry, and nursing [7,8].

In line with these changes, this study investigates KMDs’ recognition of the current state of the NLE-KMD, perceptions of the items that need to be urgently improved, and thoughts on the items that must be emphasized in NLE-KMD development in terms of overall and occupational competency. These results will serve as an essential reference for examination improvement.

## 2. Materials and Methods

### 2.1. Study Design

This study’s design recognizes KMDs’ opinions on NLE-KMD. The survey contained 16 questions with the following sections; (1) consent to voluntary participation, (2) characteristics of the KMD respondent, (3) the recognition of the current NLE-KMD, (4) the perception of what needs improvement, (5) what needs emphasis, and (6) the agreement of clinical occupational competency-related items.

We developed the initial draft using literature research and focus group interviews (FGI). For the literature study, we targeted previous and related studies on the NLE-KMD using the following databases: MEDLINE, KoreaMed, the Korean Medical Database, the Korean Studies Information Service System, ScienceOn, Korea Institute of Science and Technology Information, and the Oriental Medicine Advanced Searching Integrated System.

We conducted FGI to clarify the draft’s direction. As a result, seven KMDs who experienced the NLE-KMD within the last five years and are currently in charge of KM education participated as the representative of each KM subject. These subjects included acupuncture and moxibustion medicine, neuropsychiatry of Korean medicine, ophthalmology and otorhinolaryngology and dermatology of Korean medicine, internal medicine of Korean medicine, pediatrics of Korean medicine, rehabilitation medicine of Korean medicine, and SCM. Through this process, we identified and derived problems in KM education.

We created the actual draft through discussion with internal experts. Four researchers (EJK, HBC, JSH and YSK) held five meetings. As a result, they confirmed: (1) whether the KMD is aware of the current state of the national licensing examination, (2) which items KMDs want improved first, (3) which items KMDs want emphasized in the NLE-KMD, and (4) which clinical occupational competencies KMDs considered important from a future-oriented perspective.

After that, several meetings and subsequent revisions by external researchers (six professors at KM college and three researchers at the Institute of Korean Medicine Education and Evaluation) developed a draft. Lastly, external experts and the panel completed the final draft.

### 2.2. Participants and Procedures

We prepared a web version of the questionnaire to ensure complete answers, prevent omissions, and facilitate statistical analysis. We set an algorithm that sounded an alarm when there was an omission and prevented the respondent from moving on to the next question if the respondent did not complete the answer. This cross-sectional survey targeted participants who (1) were qualified to take the NLE-KMD after completing the official education pathway and (2) passed the NLE-KMD and obtained a KMD license. Those who qualified to take the examination but never obtained a KMD license were excluded. After obtaining a list of subjects who satisfied these criteria from the Association of Korean Medicine, we sent an e-mail to 22,338 KMDs twice at an interval of one week (22 and 28 February 2022) under the association’s approval. There was no minimum sample size due to the nature of the survey, but we tried to recruit as many respondents as possible. In addition, we gathered survey responses using the “Moaform” online survey platform from 22 February to 4 March 2022. There were no conflicts of interest in sending the e-mail or collecting the survey data.

### 2.3. Ethics

This study was approved by the Institutional Review Board of Dongguk University Bundang Oriental Hospital (DUBOH 2022-0002). When we sent the e-mail, we provided the following information: the purpose of the survey, the survey response method, required time, the e-mail address to which inquiries could be sent, the data collection method, and the academic use of survey results. We also informed the KMDs of the survey’s confidentiality regarding personal information and their right to withdraw from the study. Before the KMDs responded to the survey, we rechecked their voluntary participation by asking them to confirm their consent in the first question.

### 2.4. Statistical Methods

The online survey platform provided the raw data in Microsoft Excel format, and the researcher checked the number of responses to each question. The categorical variables included frequencies and percentages, while Likert scale variables were converted into continuous variables (i.e., five scales into 1–5 points) and presented as means ± standard deviations. Considering the statistical significance in the subgroup analysis and response proportion, we divided the participants into two subgroups based on the license acquisition period of five years. To compare the two groups, we used the Chi-square test for categorical variables and the Wilcoxon rank sum test or two-sample *t*-test for continuous variables. We assumed significance when *p* value < 0.05 using STATA version 15.0 (STATA Corp, LP. College Station, TX, USA). Lastly, one independent researcher reviewed these data repeatedly.

## 3. Results

### 3.1. Participants

Among the 1281 online questionnaire users, 1244 respondents (97.1%) agreed to voluntary participation. This stat means 5.56% of the response rate among total KMDs (22,338). Regarding the license acquisition year, 30.47% (*n* = 379), the majority, acquired their licenses within the last five years. In the other fields, the relative majority had clinical experience within five years (*n* = 483, 38.83%) and worked at a KM clinic (*n* = 648, 62.49%) (Table 1). For this reason, we divided two subgroups based on the license acquisition period of five years.

### 3.2. Recognition of the Current NLE-KMD

We measured KMDs’ awareness of the NLE-KMD update on a four-point scale (strongly agree/agree/disagree/strongly disagree) and counted the number of strongly agree and agree. Among 1244 respondents, 721 KMDs (57.95%) recognized that the NLE-KMD gradually changed over the past decade. In particular, KMDs with ≤5 years of license acquisition (UFLA) showed remarkable awareness, with 89.45% (*n* = 339) strongly agreeing or agreeing compared to 44.16% (*n* = 382) of KMDs with >5 years of license acquisition (OFLA) (Table 2).

### 3.3. Items for Improvement in the NLE-KMD

We investigated the NLE-KMD items that KMDs thought should be improved first. We provided eight items and asked respondents to select their top four priorities. Among them, “Focus on Korean standard classification of diseases (KCD) that are critical and highly frequent in clinical practice” was ranked first (*n* = 890, 18.53%), followed by “Focus on the clinical tasks, not subjects” (*n* = 767, 15.97%) and “The introduction of clinical skills tests to evaluate the clinical work performance” (*n* = 734, 15.28%).

Regarding the subgroup analysis, there was a large difference between the two groups on “The introduction of clinical skill tests to evaluate the clinical work performance”, with the UFLA group ranking this item first (18.36%) and the OFLA group ranking it fourth (13.94%) (Table 3).

### 3.4. Items for Emphasis in the NLE-KMD (Overall)

We asked the participants about the NLE-KMD items they would emphasize considering the job skill level of the KMDs who recently obtained a license. We gave the KMDs nine items with a 10-point Likert scale and compared each item by calculating the average.

As a result, the first- and second-ranked items were all related to the KCD. The most emphasized item was “The knowledge and skills related to the assessment and diagnosis of diseases (KCD)” with 8.25 points, followed by “The knowledge and skills regarding the diseases (KCD) frequently treated at primary healthcare institutes” with 8.15 points. Both subgroups gave these items high scores.

In the subgroup analysis, we identified significant differences between the two groups in the nine items, with the largest being “The knowledge and its’ application of traditional KM theories”. The OFLA group rated the item 6.12 points, while the UFLA group rated it only 4.63 points (*p* < 0.001) (Table 4).

### 3.5. Items for Emphasis in the NLE-KMD (Related to Occupational Competency)

In this section, we proposed a clinical occupational competency-related NLE-KMD system consisting of three major categories (preventive and public health activities, diagnosis and treatment of disease, and management of treatment methods). In addition, we suggested two or three items within each major category and provide a detailed explanation for each item.

Then, we asked the KMDs for agreement on (1) this proposed system and (2) each item in the three major categories in a future-oriented perspective with a five-point Likert scale and calculated the agreement by average. While overall consent was 3.75 out of 5, with insignificant differences between the two subgroups, we investigated the agreement of each item in the three major categories.

In the preventive and public health activities section, three items ranged from 3.70 to 3.93. In the subgroup, there was a difference in disease prevention and health promotion activities (*p* = 0.007).

In the diagnosis and treatment of disease section, two items (diagnosis and treatment of symptomatic disease, diagnosis and treatment based on KCD) showed a high level of agreement (>4.3). In the subgroup, we found that the OFLA group emphasized the Sasang constitution-based diagnosis and treatment while the UFLA group valued KCD (*p* < 0.001).

The area of management of treatment methods consisted of two items (management of acupuncture and other medical devices, management of herbal medications). The management of herbal medications item showed a significant difference between the two groups (Table 5).

### 3.6. Keywords for Subjective Responses

To understand individual opinions, we provided an opportunity to give subjective answers. We analyzed 558 subjective responses by setting keywords and calculating their frequency. The most referenced keywords were Sasang constitution (*n* = 252), clinical trials (*n* = 243), and prevention (*n* = 193). The combination of keywords related to the words analyzed with frequency is as follows: Participants related diagnosis to terms such as clinical, KCD, competency, and diagnostic devices. In the SCM category, respondents highlighted words such as objective, base, and standard, and simultaneously selected conflicting words such as unique and characteristics. Regarding prevention, keywords included public health, responsibility, health promotion, and COVID-19 (Figure 1).

## 4. Discussion

KM is one type of complementary and alternative medicine (CAM) that shares aspects of Chinese and Japanese medicine [9], and it is recognized as an essential contributor to CAM development [10]. The representative academic difference between KM and TCM is the existence of SCM. SCM is characterized by the quadrant diagram, including four major organs (lungs, spleen, liver, and kidneys), and the theory that structural and functional changes occur due to deviations in constitutional properties. Through this, it emphasizes individual idiosyncrasies [11]. KM was the first CAM to be covered by national health insurance, and it became an axis of the national health system of South Korea [12]. With patients’ diverse needs and the growing demand for CAM [13,14], the importance of KM is emerging. Thus, KM college students should take the required education course to qualify for the NLE-KMD. While the KHPLEI continues to address NLE-KMD challenges and developments [15], we conducted our survey to refer to the recognition and perception of existing KMDs more systematically and scientifically.

The results of Table 3 and Table 4 show that most KMDs defended the changes based on clinical practice rather than KM theory. This result is consistent with recent studies focusing on clinical practice. According to Dewey, a pioneer of experiential learning, learners can achieve proper education when it is connected to or integrated into their experience, and they can learn by reflecting on their experiences [16]. Im contends that clinical practice is an essential part of basic medical education, allowing students to acquire crucial clinical and communication skills necessary for patient treatment based on the knowledge learned in related lectures. Therefore, students can seek help in preparing for practice, clinical professors can provide opportunities and feedback for students to participate deeply in treatment, and operating institutions can design and support clinical practice curricula and resources so that students can achieve their goals while actively practicing [17].

In this context, the Institute of Korean Medicine Education and Evaluation reorganized the written tests to examine applicants’ abilities in various areas. The KM education accreditation standards were published in 2022, and they emphasized active clinical practice [18]. They also introduced the objective structured clinical examination (OSCE) and clinical practice examination (CPX) to test students’ clinical abilities. This addition enabled the competency evaluation of Korean medical school examinees and improved the education system [6]. In recent years, 12 KM colleges in Korea were also striving to improve education and evaluation. Among them, Pusan National University dramatically upgraded its KM education by introducing the following: clinical symptoms study courses integrating diverse clinical specialties, problem-based learning, the OSCE, the CPX, clinical skills tests, training and testing with standardized patients, etc. [19]. As such, the results of this survey suggest the need for a competency evaluation method and education system for examinees to improve clinical performance and professionalism.

This trend can also be seen in other countries. German medical education consists of a two-year preclinical segment and a four-year clinical segment similar to Korea [20], and the current trend is to favor education in practical settings rather than theory-based lectures [21]. Although the medical education courses in the USA differ from those in Germany, its occupational competency-centered trends are similar [20]. Since the need for a curriculum type that more effectively solves patients’ problems was suggested in 2001 [22], one study interpreted the various changes in American medical education as reflecting the intention of focusing on skills rather than knowledge [23]. Looking at the previous studies, competency-based medical education might be a method that has a positive effect on patients and communities [24].

Another notable result was the absolute agreement of the disease approach based on the standard classification of disease. Korea developed and stipulated the KCD, based on the International Classification of Diseases (ICD) and related health problems, to capture the clinical picture and produce statistics on KM [9]. After continuous revisions, Korea published the KM volume of the KCD in 1973. In 2010, the third edition of the Korean Classification of Diseases of Oriental Medicine (KCDOM3) was incorporated into the Korean modification of the ICD-10, or KCD 6, using U codes (U20–U99) [25]. In this aspect, the KCD 6 was groundbreaking, as it was the first publication in which Western medicine and traditional medicine shared a common platform [26]. In Korea, KMDs are advised to use the KCD 6, based on Western medicine, as their primary code system; however, when doctors cannot correlate a diagnosis specifically to the KCD 6, they are to supplement the diagnosis with a U code [9].

For this reason, our survey used KCD-related items, including knowledge and skills, differential diagnoses, and diagnosis and treatment. Each item received a high score of 8 or more out of 10. The KCD ranks outpatients as follows: first are those with diseases of the musculoskeletal system (52.44%), second are those with injury and poisoning (19.56%), third are those with diseases with KM names not matched with a Western medicine disease classification (7.60%), fourth are those with diseases of the nervous system (4.43%), and lastly are those with diseases of the digestive system (4.24%) [12]. Based on this, our survey’s results indicate that the current tests and education do not sufficiently reflect frequently seen KCD diseases and thus require improvement.

At the same time, it is difficult to code a diagnosis when there is little agreement on its reasons. For instance, Hua et al. [27] asked TCM practitioners to diagnose knee osteoarthritis using four TCM diagnostic methods (inquiry, (visual) inspection, auscultation/olfaction, and palpation) and found low agreement using inspection and palpation parameters. In addition, there are no codes for the diseases unique to KM, such as blood stasis syndrome and cold hypersensitivity of the hands and feet [28,29]. Therefore, to solve these problems, a more objective tool to understand and utilize each disease must be developed. In addition, improving the quality of and access to reasonable primary medical care should be possible by moving to evidence-based medicine through clinical practice guidelines (CPGs).

TCM studies also confirmed the above direction of KM pursued by KMDs. For example, Lam et al. [30] analyzed the impact of the TCM inclusion in the ICD-11 and expected that the ICD-11 would trigger TCM to provide standardized terminology, develop the CPGs, and expand education. However, Ren et al. [31] reported that only 12% of previous Western medicine guidelines in China included TCM and treated it as a low recommendation. Therefore, the authors considered that the popularity of TCM was not reflected fully due to inadequate evidence and conducting of relevant studies, systematic reviews, and meta-analyses. Finally, regarding modernization, Xu et al. [32] summarized the history of change in TCM from 1950 to 2012 and advised that TCM needed to achieve integrity, integration, and innovation to play a more significant part in future medicine.

Our study also showed two interesting results. First, there was a distinct difference in participants according to the license acquisition period. The OFLA group focused on CPG usage, new health technology, traditional theory, and other CAM. In contrast, the UFLA group emphasized clinical examination, CBT, and diagnosis and treatment based on the KCD. We could interpret this difference as the younger group being interested in the current problems they face in the field. It also suggests evaluating integrated clinical competencies beyond written tests limited to factual knowledge. This interpretation is in line with Yim’s evaluation, including that a greater proportion of problem-solving items reflects the practical situation that medical graduates face in the field. In addition, introducing case studies helps measure the ability of examinees to cope with real-world problems [6].

On the other hand, the older group is interested in applying a wide range of techniques and a practical approach to the clinical field. The generation gap in the medical field was described in several articles as a “lack of professionalism” [33] or “failure of instruction from the older generation (the Silent Generation or Baby Boomers) to the younger generation (Generation X or Millennials)” [34]. The differences between generations are likely to result from the characteristics of the sociocultural environment and the influence of other elements characterizing an era (such as industrialization and democratization) [35]. It was reported that the older generation tends to “respect the system” while the younger generation tends to “respect expertise” [36], which is in line with our results. Understanding intergenerational differences was described as an essential task.

Second, our study allowed respondents to present their opinions in subjective responses so we could understand their thoughts in depth. In these opinions, we found frequent mentions of public healthcare, such as KMDs playing a role in COVID-19. Several studies reported the effect of KM on COVID-19, but KMDs cannot play an active role due to institutional reasons. These mentions are related to the desire to expand the social role of KMDs to evaluate social competencies [37,38,39].

On the topic of the NLE-KMD, to our knowledge, our study is the latest to investigate and analyze the current perceptions of KMDs and the first to gather more than 1000 KMD opinions. Because of our survey’s significant number of respondents, our study potentially provides a more representative justification than previous surveys in which KMDs participated [40]. However, there are some limitations. First, as we conducted this survey using e-mail, it is unlikely that we guaranteed access to all KMDs. Second, more than 1000 KMDs participated in this survey, but the sample was not equally distributed in terms of age or license acquisition period. Therefore, this study does not represent the opinions of all KMDs. Third, we did not include the opinions of other specialists, such as pedagogists.

Nevertheless, the data can facilitate the development of a national examination. As previous studies revealed, evaluating clinical occupational competency based on core competencies in the framework of disciplinary education is not easy. Therefore, based on the perceptions of the KMDs in this study, it is necessary to establish a long-term plan for the systematic flow between the curriculum and evaluation by researching detailed evaluation standards and practices to apply to KMDs with clinical occupational competency.

## 5. Conclusions

This was the first survey to officially investigate the perceptions of KMDs on the NLE-KMD. More than 1000 KMDs participated, and we were able to identify the items KMDs wanted to improve. Moreover, they emphasized (1) clinical practice including clinical tasks and clinical work performance and (2) KCD-related items. Regarding the KCD, KCD diseases frequently seen in clinical practice and related diagnostic knowledge and skills were considered important. Although there were some discrepancies according to the year of license acquisition and limitations, these results suggest an emphasis on clinical occupational competency.

It is hoped that this study induces future research on educational subjects. This survey was conducted with KMDs who already passed the NLE-KMD, and they were not directly affected by the current KM education. Thus, a further survey on the perceptions of KM college students, as educated subjects, is needed. Each KM college has an academic committee established by students, and through various types of research involving this population, including qualitative research or surveys, a more comprehensive direction of the NLE-KMD should be established. Moreover, an investigation into the perceptions of professors should be conducted. This survey investigated only the general direction and did not check the opinions on each KM college subject. However, this process would help set the direction and development of the future NLE-KMD.

## Figures and Tables

**Figure 1 healthcare-11-01685-f001:**
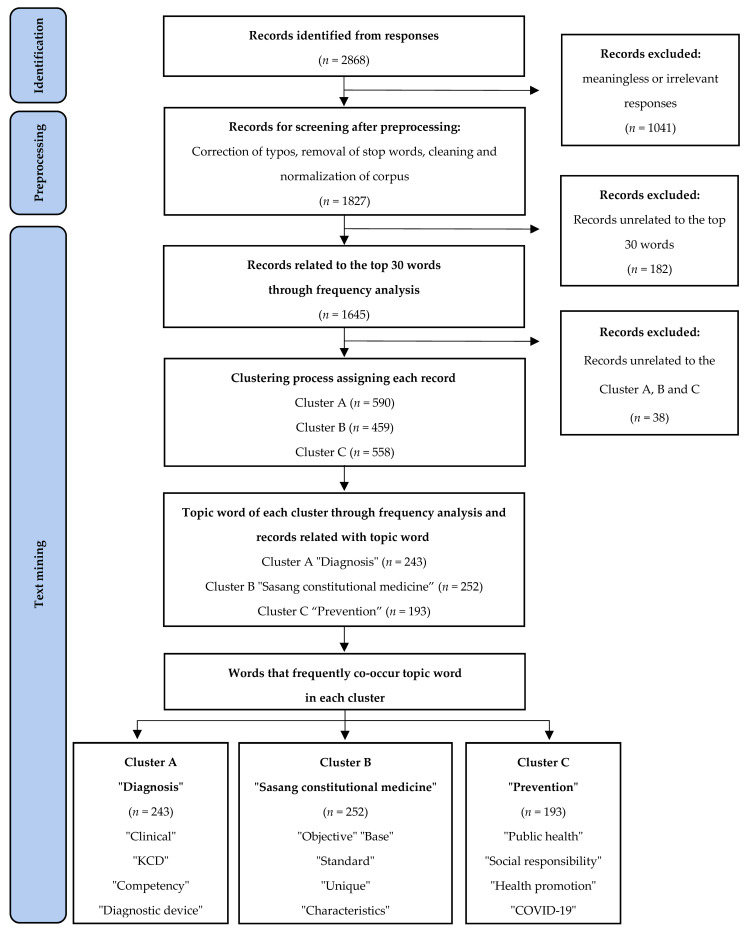
The process of clustering keyword according to frequency.

**Table 1 healthcare-11-01685-t001:** Characteristics of respondents.

Question	Classification		*n* (%)
The acquisition year of Korean medicine doctor license	2018–2022		379 (30.47)
	2013–2017		267 (21.46)
	2003–2012		336 (27.01)
	<2003		262 (21.06)
	Total		1244 (99.9)
The year of clinical experience of Korean medicine	0		83 (6.67)
	≤5		400 (32.15)
	5 < x ≤ 10		261 (20.98)
	10 < x ≤ 20		316 (25.40)
	20<		184 (14.79)
	Total		1244 (99.9)
Whether working at educational institution	No	Clinical, clinics (director of the clinic or assistant director of the clinic)	648 (62.49)
		Clinical, hospitals (director, paid-doctor, or specialist at Korean medicine hospital, convalescent hospital, or medical center)	182 (17.55)
		Clinical, public health doctor or military doctor	145 (13.98)
		Non-clinical (alternative military service, personal business, parenting, studying other fields)	44 (4.24)
		Others	18 (1.74)
		Total	1037 (83.36)
	Yes	Clinical, hospital affiliated with college of Korean medicine (resident, fellow, professor, others)	169 (81.64)
		Non-clinical, college of Korean medicine (teaching assistant, researcher, professor, others)	38 (18.36)
		Total	207 (16.64)
	Total		1244 (100.0)

**Table 2 healthcare-11-01685-t002:** The awareness of the recent changes in the NLE-KMD.

Question	Classification	OFLA *n* (%)	UFLA *n* (%)	Total (OFLA + UFLA) *n* (%)	*p* Value
Whether recognizing the recent changes in the NLE-KMD	Strongly agree	80 (9.25)	114 (30.08)	194 (15.59)	<0.001 *
	Agree	302 (34.91)	225 (59.37)	527 (42.36)	
	Disagree	342 (39.54)	36 (9.50)	378 (30.39)	
	Strongly disagree	141 (16.30)	4 (1.05)	145 (11.66)	
	Total	865 (100.0)	379 (100.0)	1244 (100.0)	

NLE-KMD: national licensing examination for Korean medicine doctors, OFLA: >5 years of license acquisition, and UFLA: ≤5 years of license acquisition. * Chi-square test.

**Table 3 healthcare-11-01685-t003:** The items to be improved in the NLE-KMD.

Question	Classification	OFLA *n* (%)	UFLA *n* (%)	Total (OFLA + UFLA) *n* (%)	*p* Value
Which items should be improved regarding the current NLE-KMD? (Please select within 4 items to prioritize)	Non-essential or memorization questions should be removed from the question bank.	484 (14.45)	185 (12.72)	669 (13.93)	<0.001 *
	The scope of the various subjects with overlapping questions should be readjusted.	292 (8.72)	136 (9.35)	428 (8.91)	
	New questions should be updated to reflect a new medical technology.	319 (9.53)	96 (6.60)	415 (8.64)	
	New questions should be developed based on the Korean medicine clinical practice guideline.	423 (12.63)	180 (12.38)	603 (12.55)	
	The scope of the examination should be focused on the clinical tasks, not subjects.	538 (16.06)	229 (15.75)	767 (15.97)	
	The scope of the examination should focus on disease (KCD) that are critical and highly frequent in clinical practice.	630 (18.81)	260 (17.88)	890 (18.53)	
	Clinical skills test should be introduced to evaluate the clinical work performance.	467 (13.94)	267 (18.36)	734 (15.28)	
	A variety of forms of new questions should be developed to be used on a computer-based test (CBT).	196 (5.85)	101 (6.95)	297 (6.18)	
	Total	3349 (100.0)	1454 (100.0)	4803 (100.0)	

NLE-KMD: national licensing examination for Korean medicine doctors, OFLA: >5 years of license acquisition, UFLA: ≤5 years of license acquisition, and KCD: Korean standard classification of disease. * Chi-square test.

**Table 4 healthcare-11-01685-t004:** The items to be emphasized in the NLE-KMD.

Question	Classification	OFLA (*n* = 865)	UFLA (*n* = 379)	Total (OFLA + UFLA) (*n* = 1244)	*p* Value
Which items should be emphasized first in the NLE-KMD? (Please check with a 10-point Likert scale)	Knowledge and its’ application of traditional Korean medicine theories.	6.12 ± 2.64	4.63 ± 2.51	5.66 ± 2.69	<0.001 *
	Knowledge and its’ application of biomedical sciences.	6.81 ± 2.21	6.79 ± 2.24	6.80 ± 2.22	0.868
	Knowledge and communication skill to collaborate with various public health experts.	7.36 ± 2.17	7.76 ± 2.11	7.48 ± 2.16	0.001 *
	Knowledge and skills in conventional treatments such as acupuncture, moxibustion, and herbal medicines.	7.83 ± 2.14	7.93 ± 2.05	7.86 ± 2.12	0.499
	Knowledge and skills of medical technology such as pharmacopuncture, chuna, thread-embedding therapy, and new herbal medicinal preparations (e.g., new pharmaceutical preparations).	7.30 ± 2.23	7.72 ± 2.23	7.43 ± 2.24	0.001 *
	Knowledge and skills in various overseas traditional medicine or complementary and alternative medicine.	5.19 ± 2.34	4.56 ± 2.39	5.00 ± 2.37	<0.001 **
	Knowledge and application of evidence-based Korean medicine (e.g., medical articles).	6.97 ± 2.28	7.31 ± 2.26	7.07 ± 2.28	0.008 *
	Knowledge and skills regarding the disease (KCD) frequently treated at primary healthcare institutes.	8.06 ± 2.05	8.36 ± 1.97	8.15 ± 2.03	0.006 *
	Knowledge and skills related to assessment and diagnosis of diseases (KCD).	8.15 ± 2.05	8.48 ± 2.04	8.25 ± 2.06	0.001 *

NLE-KMD: national licensing examination for Korean medicine doctors, OFLA: >5 years of license acquisition, UFLA: ≤5 years of license acquisition, and KCD: Korean standard classification of disease. * Wilcox rank sum test. ** Two-sample *t*-test.

**Table 5 healthcare-11-01685-t005:** The items to be emphasized in the NLE-KMD related to occupational competency.

Question		Classification	OFLA (*n* = 865)	UFLA (*n* = 379)	Total (OFLA + UFLA) (*n* = 1244)	*p* Value
Overall (three categories)		Degree of agreement on the following three proposed system (preventive and public health activities/diagnosis and treatment of disease/management of treatment methods).	3.74 ± 0.84	3.76 ± 0.82	3.75 ± 0.84	0.868
Each three categories	Preventive and public health activities	Disease prevention and health promotion activities.	3.98 ± 0.96	3.83 ± 0.97	3.93 ± 0.97	0.007 *
		Carrying out social responsibilities.	3.70 ± 1.00	3.67 ± 0.99	3.69 ± 1.00	0.540
		Public health management.	3.66 ± 1.03	3.78 ± 0.98	3.70 ± 1.01	0.112
	Diagnosis and treatment of disease	Diagnosis and treatment of symptomatic disease.	4.36 ± 0.81	4.31 ± 0.88	4.34 ± 0.83	0.612
		Diagnosis and treatment based on Sasang constitution.	3.09 ± 1.11	2.77 ± 1.13	2.99 ± 1.12	<0.001 **
		Diagnosis and treatment based on KCD.	4.32 ± 0.84	4.50 ± 0.78	4.37 ± 0.83	<0.001 *
	Management of treatment methods	Management of acupuncture and other medical devices.	4.10 ± 0.98	4.15 ± 0.99	4.11 ± 0.99	0.246
		Management of herbal medications (including medicinal herbs).	3.98 ± 1.01	3.78 ± 1.08	3.92 ± 1.04	0.003 *

NLE-KMD: national licensing examination for Korean medicine doctors, OFLA: >5 years of license acquisition, and UFLA: ≤5 years of license acquisition, KCD: Korean standard classification of disease. * Wilcox rank sum test. ** Two-sample *t*-test.

## Data Availability

Data are available from the corresponding author upon reasonable request.

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
