# Peer review of "A Survey on Perceptions of the Direction of Korean Medicine Education and National Licensing Examination"

_healthcare, 2023, doi:10.3390/healthcare11121685_

Round 1
Reviewer 1 Report
This is an interesting topic for traditional medicine education in Korea.
As the authors noted in limitations, even if authors could not collect the any opinions of other pedagogists, it will be possible to perform comparative analysis by all respondents and Korean medicine doctor working in Clinical, hospital affiliated with college of Korean medicine who have been licensed several years (e.g. 5years) ago as experts.
In line 235, the phrase 'Im said' needs to be corrected.
Reviewer 2 Report
The paper presents a high quality of processing in terms of methodology, contents and results. As regards the introduction, it would be appropriate to increase the number of references and the length of the introduction, also introducing the background paragraph to provide a broader overview of the existing scientific literature also through a rapid revision of the same.
In particular for medical education, today, is important the challenge of diversity in health. At this regard I suggest to include in the paper and to study the following paper: https://doi.org/10.13136/isr.v13i1.589
From a methodological point of view, I recommend providing the inclusion and exclusion criteria for recruited subjects more explicit because it would give further added value to this high-quality work.
The results are well represented both graphically and from a descriptive point of view, however,
The discussion is well structured, but I would suggest: 1) to increase the references because they are very few compared to the contents reported; 2) the conclusions must also be better specified by writing a paragraph entitled conclusions, making known what future research developments may be.
The conclusions are consistent with the evidence and arguments presentedand address the main question asked, but should be placed in a separate paragraph from the discussion and entitled "Conclusions".
Extensive editing of English language required
Reviewer 3 Report
Overview and general recommendation
This study focuses on the opinions and perceptions of medical professionals who have taken the National Licensing Examination in South Korea about the National Licensing Examination. There is a conflict about which subjects it should assess, as well as its methodological model of assessment. My main recommendation is that, as this is a very country-focused study, the results obtained should be compared, in the Discussion, with the methods and areas of knowledge with which they are assessed in other countries, not only through the approximation with the closest countries discussed in the Introduction. In fact, such a national examination system is used for medical specialities in other countries, especially in North America and Europe.
Major comments
On one hand, the Study Design sub-section does not mention the type of study design (cross-sectional study). Also, although it is not mandatory, it is advisable to follow the STROBE principles for cross-sectional studies to ensure that most of the checklist items are met.
As for the survey, although it is not a validation, no data have been shown to support its consistency (at least Cronbach's alpha values, for each dimension and the whole measurement instrument).
On the other hand, much data appear in Results, mainly in the legends of the tables and at the beginning of the sub-sections, which should go to Methodology. For example, the explanations of the measurement instruments and the bivariate statistical tests carried out.
Finally, Table 1 focuses on demographic characteristics, but there is no real demographic variable (sex, age,...). There are two options: either change the title or add the demographic variables if they were collected.
Minor comments
- Lines 53-58: How is the exam currently structured, and how is it assessed?
- Line 113: There is a double space on this line.
- Line 124: The comment referring to another study should be moved to Discussion.
- Line 125: It is confusing to mention "majority". Please replace it with "relative majority".
- Lines 125-127: I think this part could be commented on as the main profile of the participants who responded to the survey.
- Lines 127-128: This last sentence should be in Methodology.
- Table 1: (1) Please insert an "x" between the "< ≤" symbols. (2) The sum of the percentages in the second question is 99.9%. (3) Remember to add the comma in the thousands.
- Tables 2-5: (1) Is there a need for a column to join the data of both groups of participants when you are comparing them? (2) Please do not include more than three decimal places for the p-value. In case of smaller values, express it as "p<0.001". (3) When dealing with participants in subgroups, the number of subgroups should be expressed as "n" and not as "N".
- Figure 1: (1) The heading of Figure 1 does not appear in Figure 1. (2) From the middle of the flowchart onwards, Cluster C and its numbers are not reflected. (3) Please include the frequencies of each keyword you have included in each cluster at the end of the flowchart.
- Lines 228-232: Please do not refer to Results data that has already been discussed.
I hope my comments will help you to improve the manuscript.
Best regards.
Reviewer 4 Report
Dear authors,
I have reviewed your article on the awareness and opinions of Korean physicians regarding updates to the Korea Health Personnel Licensing Examination Institute. While the analysis appears to be well-conducted, I believe that some improvements can be made. Please see the list below for my suggestions:
Abstract: Please rewrite the abstract in a more straightforward manner, limiting the use of acronyms and providing more space to explain the main goals of the article.
Introduction: Please address your research question more deeply and clearly in the introduction.
General Comment: Can you provide insights into the opinions of the international medical community regarding Korean traditional medicine?
Line 95: How can any questionnaire guarantee complete answers or prevent omissions? Please clarify what you meant.
Line 112: Your sentence about minimizing missing data is not clear.
Line 131: Did you only ask if respondents were updated about NLE-KMD without actually verifying it?
Lines 150-153: You present an interesting result, but you did not discuss its implications. Please discuss your results throughout the article.
Lines 167-170: You mention an apparent generational gap, but you should discuss this in more depth.
Section 3.5: You discuss differences but not their implications. Please explore the question of "Who prefers what?"
Line 192: The word "Sasang" appears out of nowhere. Please explain its meaning.
Discussion: Please move theoretical issues to the introduction and focus your discussion on your findings and their implications, relating them to the concepts presented at the beginning.
Thank you for considering my suggestions. I look forward to seeing your revised manuscript.
Best regards
Dear authors,
I regret to inform you that the manuscript requires a thorough review by a proficient English speaker. There are several typos and some sentences that are difficult to understand. Furthermore, I have noticed that you have employed some "false friends," where the intended meaning is unclear. For instance, the phrase "KMD's recognition" is ambiguous.
Another issue is the extensive use of acronyms throughout the article. The number of acronyms is excessive, and they are not listed in a legend, making it challenging for readers to understand the manuscript. This places a significant burden on readers, and it would be helpful to reduce the number of acronyms or provide a legend to explain them.
Thank you for your cooperation in addressing these issues.
Round 2
Reviewer 3 Report
Dear authors,
Having checked the changes you have made, I consider that you have made the proposed amendments.
Best regards.
Reviewer 4 Report
Dear authors,
Thank you for your response. In my opinion, you have done a good job with your manuscript, effectively improving it and making it suitable for publication.
I would like to add a couple of considerations, and it is up to you whether or not to follow them:
I still believe that your abstract could benefit from being rewritten to provide less information, but more focused and important details. In my opinion, the abstract should not aim to summarize the entire article, but rather to provide a few important concepts that make it immediately clear to readers whether the paper is of interest to them. They can then delve deeper into the main text to explore the concepts presented.
I still feel that your introduction lacks a clear statement of the research question.
However, these are minor issues in an otherwise good article. Best regards.
